# The Effect of the Initiator/Activator/Accelerator Ratio on the Degree of Conversion, Film Thickness, Flow, and Cytotoxicity of Dual-Cured Self-Adhesive Resin Cements

**DOI:** 10.3390/ma17143572

**Published:** 2024-07-19

**Authors:** Hyun Kyung Moon, Jong-Eun Won, Jae Jun Ryu, Ji Suk Shim

**Affiliations:** 1Department of Biomedical Sciences, College of Medicine, Korea University Guro Hospital, Seoul 08308, Republic of Korea; coolmhk@korea.ac.kr; 2Institute for Clinical Dental Research, Department of Dentistry, Korea University Guro Hospital, Seoul 08308, Republic of Korea; jewon0405@gmail.com; 3Department of Prosthodontics, Korea University Anam Hospital, Seoul 02841, Republic of Korea; 4Department of Prosthodontics, Korea University Guro Hospital, Seoul 08308, Republic of Korea

**Keywords:** self-adhesive resin cements, initiator, activator, accelerator, degree of conversion, auto polymerization

## Abstract

Although self-adhesive resin cements are convenient and less technique-sensitive materials for dental clinicians, they exhibit a lower degree of conversion due to acidic components in their composition. Supplementation of the initiator, accelerator, and activator in self-adhesive resin cements has been suggested to compensate for the lower degree of conversion. This study aimed to evaluate the effects of different combinations of self-curing initiators, self-curing activators, and accelerators on the degree of conversion (DC) of self-adhesive resin cements. A dual-cured self-adhesive resin was prepared using six combinations of initiators, activators, and accelerators. The change in the DC over time was evaluated with and without light curing. The film thickness, flow properties, and cytotoxicity of each formulation were assessed. The results showed that all supplemental components had an effect on increasing the DC, but a greater increase in the DC was observed in the following order: activator, accelerator, and initiator. The cytotoxicity of the resin cements was related to the DC values, as resin cements with lower DC values exhibited higher cytotoxicity. The film thickness met the ISO standards for all groups. The results suggest that utilizing an activator is the most effective approach to enhance the DC in self-adhesive resin cement and that cytotoxicity tended to increase with lower DC values, whereas film thickness and flow properties demonstrated no correlation with DC values.

## 1. Introduction

Resin cements are used in esthetic restorations because they have superior mechanical and chemical properties compared to traditional dental cements, such as zinc phosphate, acrylic, and glass ionomer cements [1,2]. Resin cements are classified into self-cured, light-cured, and dual-cured resin cements according to their curing mechanisms. Dual-cured resin cements have been developed in a desirable way to stabilize the restoration with initial rapid polymerization under light curing, as well as to achieve complete polymerization by self-curing under restorative materials that attenuate light to cements [3,4,5]. Achieving a higher degree of conversion (DC) is important to provide better mechanical strength and avoid the unfavorable biological effects of unreacted monomers on the pulp and gingival [6]. Since the 2000s, self-adhesive resin cements that are used in a single step without pretreatment, including etching, priming, and bonding, have been introduced as alternatives to conventional resin cements, which require multiple steps [7]. Owing to the development of resin cements, including dual-cured and self-adhesive resin cements, the application of the cementation process is simple and has a lower technique sensitivity [8].

Conventional dual-cured resin cements were composed of non-acidic monomers, such as triethylene glycol dimethacrylate (TEGDMA), 1,6-bis(methacryloxy-2-ethoxycarbonylamino)-2,4,4-trimethylhexane (UDMA), 2,2-bis[p-(2′-hydroxy-3′-methacryloxypropoxy)phenylene]propane (Bis-GMA), and 2-hydroxyethyl methacrylate (HEMA), self-curing initiators, such as benzoyl peroxide (BPO), photo-initiators, such as camphorquinone (CQ) and tertiary amine, which promote the activation of self-curing initiators, such as N, N-dimethyl-4-toluidine (DMPT) [1,9]. Otherwise, dual-cured self-adhesive resin cements contain acidic monomers, such as 10-methacryloxydecyl dihydrogen phosphate (10-MDP), 2-methacryloxyethyl phenyl hydrogen phosphate (Phenyl-P), and 4-methacryloyloxyethy trimellitate anhydride (4-META), and demineralize and penetrate enamel and dentin for stable bonding with restorative materials [1,8,10]. Self-etching and self-adhesive resin cements contain acidic monomers, which may chemically interact with the amine initiator and negatively impact DC, especially during auto-polymerization [11]. To overcome the disadvantages of polymerization by acidic components, dual-cured self-adhesive resin cements contain aromatic sulfinate salts, such as sodium benzene sulfonate and sodium p-toluene sulfinate (SPTS), which act as accelerators [12]. In addition, previous studies have shown that aromatic sulfinate salts protect tertiary amines by inhibiting the reaction of the acidic monomer with tertiary amines [13,14] and that using more initiators is effective in achieving a higher DC [15]. Despite the individual effects of the components in promoting the polymerization processes, their combined effects on polymerization have not yet been studied.

Resin cements should have appropriate working times for clinical use [16]. In addition, resin cements should maintain proper film thickness and flowability during the working time [17]. These factors are related to proper seating of the restoration on the prepared tooth [18]. Improper seating of the restoration by thick film thickness causes plaque accumulation, periodontal disease, and the dissolution of cement [19,20]. Moreover, self-adhesive resin cements show worse biological effects on the surrounding tissues, as they release acidic components when they are dissolved [6]. These results suggest that providing an appropriate polymerization process by adjusting the combination of the initiator, activator, and accelerators is critical for dental cements.

This study aimed to evaluate the effects of the self-curing initiator/self-curing activator/accelerator ratio on the degree of conversion, film thickness, flow, and cytotoxicity of dual-cured self-adhesive resin cements. Dual-cured self-adhesive resin cements were prepared using six combinations of initiators, activators, and accelerators. The temporal changes in the degree of conversion (DC) were evaluated under conditions with and without light curing. The film thickness, flow properties, and cytotoxicity of each formulation were assessed. The null hypothesis tested was that the initiator to activator to accelerator ratio would not affect the DC, film thickness, flow, or cytotoxicity of dual-cured self-adhesive resin cements.

## 2. Materials and Methods

### 2.1. Composition of Experimental Self-Adhesive Resin Cements

The experimental samples of self-adhesive resin cements were composed of a paste-paste system. Pastes A and B were obtained from SPIDENT (Incheon, Republic of Korea). CQ and ethyl 4-dimethylaminobenzoate (EDMAB) were used as the photo-initiators. 2-(2-hydroxy-5-methylphenyl) benzotriazole (TP) was used as a photo-stabilizer. A BPO (self-curing initiator), DMPT (self-curing activator), and SPTS (accelerator) were used to self-cure the material. Barium glass particles were used as the fillers. The components of two cement pastes are listed in Table 1.

As shown in Table 2, we prepared six groups of self-adhesive resin cements (G1, G2, G3, G4, G5, and G6) with different initiator, activator, and accelerator compositions.

### 2.2. Degree of Conversion

Two slide glasses were prepared to measure the DC. One slide glass was taped using tape with a thickness of 0.05 mm at both ends. Self-adhesive resin cements mixed by the auto-mix tip were placed at the center of a taped slide glass and covered with another slide glass. The DC was measured in different curing modes.

**Mode 1**: Each self-adhesive resin cement in the group (*n* = 6) was self-cured in the dark at room temperature.**Mode 2**: Each self-adhesive resin cement in the group (*n* = 6) was light cured through 1 mm thickness zirconia ceramic (LAVA^TM^ Plus, 3M Deutschland GmbH, Neuss, Germany) for 30 s at room temperature. A light-curing unit (Be-Lite, B&L Biotech, Fairfax, VA, USA) with an irradiance of 1000 mW/cm^2^ was used.

The DC was measured by Fourier transform–near-infrared spectroscopy (NIRSolutions^TM^, BUCHI, Flawil, Switzerland) following a timeline of up to 2 weeks (Figure 1). The spectral range acquired 12,500–4000 cm^−1^ with a resolution of 8 cm^−1^. The DC was calculated by measuring the area of the vinyl stretching peak at 6165 cm^−1^. The initial peak area was acquired immediately after the resin cement was mixed, and the final peak area was acquired following the determined timeline. The DC was calculated as follows [21].
$$DC=\frac{1-final\;peak\;area}{initial\;peak\;area}\times 100\%$$

### 2.3. Film Thickness

The film thickness of the self-adhesive resin cement was measured according to ISO 4049 [22]. Two flat glass plates with a contact surface area of 200 mm^2^ and a thickness of 5 mm were used. After putting 0.1 mL of the resin cement in the center of the lower plate, the upper plate was placed on the cement. At 60 s after the resin cement was mixed, a force of 150 N was applied vertically and centrally to the specimen through the top plate for 180 s. The load was applied smoothly to avoid the rotation of the glass. After 10 min of mixing, the plates from the loading device were removed, and the combined thicknesses of the two glass plates and the specimen film were measured using a digital micrometer (MCD-25MX, Mitutoyo, Kawasaki, Japan). Ten specimens per group were used for the test. The mean and standard deviation of the thickness were obtained.

### 2.4. Flow Ability

The flow test of the self-adhesive resin cements was conducted according to ISO 6876 [23]. Two flat glass plates with dimensions of 50 mm × 50 mm and a thickness of 5 mm were used. A total of 0.05 mL of resin cement was placed at the center of the first glass plate. At 180 s after mixing, a second glass plate was placed on the cement. A 100 g mass was used to make a total mass on the plate of 120 g. The weight was removed 10 min after mixing, and the maximum and minimum diameters of the compressed discs of the resin cements were measured using a digital micrometer. In case of differences in diameter within 1 mm, the mean of the two diameters was recorded. If the two diameters were not less than 1 mm, the test was repeated. Ten specimens per group were used for the test. The mean and standard deviation of the thickness were obtained.

### 2.5. Cell Viability and Fluorescence Image

The specimens were prepared by filling the resin cement in a Teflon mold with a hole 10 mm in diameter and 2 mm in thickness. The specimens in each group were divided into two groups depending on the self-cure time duration: 10 (*n* = 6) and 60 min (*n* = 6). Following ISO 10993 [24], specimens were incubated in Dulbecco’s modified Eagle medium (DMEM) at a concentration of 0.2 g/mL at 37 °C for 24 h. The extracted solution was filtered by a 0.2 μm syringe filter. Human gingival fibroblast cells (HGF-1; CRL-2014, ATCC, Manassas, VA, USA) were cultured by DMEM with 10% fetal bovine serum and 1% penicillin/streptomycin at 5% CO^2^ at 37 °C. HGF-1 cells were seeded into 96-well culture plates at a density of 5 × 10^3^ cells/well. After 24 h, HGF-1 cells were incubated for 24 h with the extracted solution diluted in a 1:1 ratio. Cell viability was determined using a cell counting kit (CCK-8, Dojindo Laboratories, Kumamoto, Japan) according to the manufacturer’s protocol. The absorbance was measured at 450 nm using a microplate reader (Spectra Max 190, Molecular Devices, Sunnyvale, CA, USA). 

Rhodamine–phalloidin staining was performed to examine cell morphology. HGF-1-exposed extracted liquid was washed with phosphate-buffered saline, fixed in 4% paraformaldehyde, and permeabilized with 0.1% Triton X-100 (Sigma-Aldrich, St. Louis, MO, USA). Rhodamine–phalloidin (Thermo Fisher Scientific, Waltham, MA, USA) was used to stain F-actin, and Hoechst 33342 (Thermo Fisher Scientific, Waltham, MA, USA) was used to stain the nuclei. Cell morphology was analyzed using a fluorescence microscope (EVOS FL Auto, Thermo Fisher Scientific, Waltham, MA, USA).

### 2.6. Statistical Analysis

To compare the DC and polymerization rate of self-adhesive resin cements depending on the initiator/activator/accelerator ratio, a two-way analysis of variance was performed with Bonferroni’s post-test. For the analysis of film thickness, flow, and cytotoxicity, a one-way analysis of variance was performed using Bonferroni’s post-test. All Statistical analyses were performed using Prism (Prism5, Graphpad, San Diego, CA, USA).

## 3. Results and Analysis

### 3.1. Degree of Conversion

Figure 2 shows the DC and polymerization rates for each group as the time flow. The statistical analyses of the DC in different curing modes are shown in Table 3 and Table 4. Figure 2a shows that G6 exhibited the highest DC (61.02%), whereas G1 exhibited the lowest DC (6.75%). Compared to G1, which had an added initiator and activator, the DC value of G3 with the activator added at a double ratio was 53.01%, indicating an almost eight-fold increase. The DC values of G4 with an added accelerator and G2 with twice the number of initiators were 14.55% and 9.91%, respectively. The initial DC of G1, G2, and G4 were slower than G3, G5, and G6. The conversion rate in G5, with an initiator added at twice the ratio compared to G4, was 51.78%, similar to that in G3, representing a moderate DC. Moreover, the experimental groups with the added accelerators consistently exhibited higher DC values when the initiator-to-activator ratio was constant.

However, Figure 2b shows that all groups exhibited DC values exceeding 60% within 60 min in the self-curing mode, and the DC continued to increase for up to 2 weeks, as shown in Table 3. Polymerization rates are shown in Figure 2c. G6 exhibited the fastest polymerization rate during the early phase, whereas G1 took the longest time to reach the maximum polymerization rate. The time points at which each group showed a significant maximum polymerization rate in the DC were as follows: G6 > G5, G3 > G4 > G2 > G1.

After light curing, as shown in Figure 2d, the DC values in all groups immediately exceeded 60% and reached a maximum within 1 min. At 60 min after light curing, Figure 2e shows that all groups displayed DC values above 70%, with the DC continuing to increase over the next 2 weeks, as shown in Table 4. There were no significant differences between the groups. Figure 2f shows that the polymerization rate reached its maximum immediately in all groups.

### 3.2. Film Thickness and Flow Distance

The film thickness of the self-adhesive resin cement is shown in Figure 3a. G4 exhibited a film thickness of 20.2 ± 3.43 μm, while G1 showed a film thickness of 26.4 ± 2.37 μm. The differences between G4 and G1 were statistically significant. The film thicknesses of G2, G3, G5, and G6 were 31.5 ± 3.44 μm, 30.9 ± 3.25 μm, 29.3 ± 2.98 μm, and 29.6 ± 3.53 μm, respectively. Statistical analysis indicated no significant differences in the film thickness between G2, G3, G5, and G6; however, there were significant differences when compared to G4. However, all groups had film thicknesses below 50 μm, in accordance with the ISO 4049 standard.

The flow distance of the self-adhesive resin cement was shown in Figure 3b. The flow distance was the longest for G4 (29.59 ± 0.82 mm), followed by G1 (28.17 ± 1.07 mm). The flow distance for other groups were as follows: G2, 26.64 ± 0.88 mm; G3, 27.27 ± 0.90 mm; G5, 27.79 ± 0.87 mm; and G6, 26.74 ± 0.65 mm. Statistical analysis indicated no significant differences in the flow distance among G2, G3, G5, and G6, but a significant difference was found between G4 and G1, G2, G3, G5, and G6, as well as between G1 and G2 and between G1 and G6.

### 3.3. Cell Viability and Fluorescence Image

Figure 4 shows the viability of HGFs after 24 h. When the self-adhesive resin cements were self-cured for 10 min, cell viability in G1, G2, and G4 was relatively lower at 11.52%, 22.30%, and 8.92%, respectively, compared to the others (see Figure 4a). The differences in G1 and G4 were significant compared with those in the other groups, while the cells cultured in G3, G5, and G6 showed higher viabilities of 86.61%, 75.12%, and 81.95%, respectively. When self-adhesive resin cements were self-cured for 60 min, cell viability in all groups exceeded 70% (Figure 4b). Cell viability in G6 was the highest at 90%, which was significantly different compared to that in G1, G2, G4, and G5.

As shown in Figure 5a, the cell morphologies in the self-adhesive resin cements self-cured for 10 min showed lower cell densities and damaged membranes in G1, G2, and G4. In contrast, a higher cell density and widely elongated morphology were observed in G3, G5, and G6. The cell morphologies of the self-adhesive resin cements self-cured for 60 min are shown in Figure 5b. In all groups, there was no effect on the organization of the F-actin network in the cellular cytosol.

## 4. Discussion

Self-adhesive resin cements are convenient and less technique-sensitive materials for dental clinicians. Owing to the acidic components in their composition, self-adhesive resin cements have shown a lower DC in their auto-polymerization in previous studies [11,25,26]. To overcome the unfavorable clinical results derived from a low DC, supplementing initiators, accelerators, and activators in self-adhesive resin cements has been suggested to increase the DC during auto-polymerization [13,14,15]. However, no study has evaluated the combined effect of these components on the DC in self-adhesive resin cement. In this study, the effects of different combinations of initiators, activators, and accelerators on the DC of self-adhesive resin cement were evaluated. Additionally, the effects of the components on film thickness, flowability, and cytotoxicity were measured to estimate the clinical performance of the cements. The results showed that all supplemental components had an effect on increasing the DC, but a greater increase in the DC was observed in the following order: activator, accelerator, and initiator. Therefore, the hypotheses of this study were rejected. The cytotoxicity of the resin cements was related to the DC values, as resin cements with lower DC values exhibited higher cytotoxicity. The film thickness met the ISO standards for all groups.

Comparing the time point at which significant changes in the DC were observed in each group, it is possible to evaluate which supplemented components more efficiently contributed to the increase in the DC, and the results in self-curing mode show that faster auto-polymerization occurred in the groups in the following order: G6, 3, 5, 4, 2, and 1. The DC of G2, 3, and 4 was significantly higher than that of G1, suggesting that all the supplementing components, including the initiator, activator, and accelerator, were effective in increasing the DC when auto-polymerizing. However, the fact that a higher DC was shown in the order of G3, 4, and 2 means that more efficient results were derived in the DC by supplementing components in the order of the activator, accelerator, and initiator. Previous studies have suggested that self-adhesive cements show a low DC in auto-polymerization because of the inhibition of the reaction of the initiator by the acidic components in the cements [11,25]. The results of this study suggest that the recovery of the reaction of the initiator by an activator is the most effective way to increase the DC in self-adhesive cement. As shown by the reaction mechanism in Figure 6, the accelerator indirectly activates the initiator by inhibiting the reaction in which the activator is suppressed by the acid monomer. The supplementing initiator also showed a positive effect on increasing the DC, but it seems to be less effective than activators or accelerators [27]. The result that the DC increased with time until 2 weeks in the light-curing mode means that auto-polymerization was prolonged for 2 weeks, although photo-polymerization was completed, which is consistent with the results of previous studies [28,29]. However, there were no significant differences in the DC between the groups in the light-curing mode, and the results showed that the differences in the combination of components only affected the DC in auto-polymerization. Previous studies have shown that self-cured specimens demonstrate a lower DC in the case of light attenuation in self-adhesive resin cements [11,30]. However, the results of this study show that the DC achieved by self-curing can vary, depending on the initiator, accelerator, and activator components.

Achieving sufficient initial polymerization in the dual-cured resin cement is necessary before the restoration is exposed to masticatory force, food, or saliva, which may lead to cement resorption and degradation [31]. Although dual-cured resin cement can be polymerized by light curing and self curing, sufficient light curing may be difficult to achieve in resin cement, depending on the attenuation effect of restorative materials [32]. Therefore, faster auto-polymerization in dual-cured resins is beneficial for overcoming the attenuation effect. Otherwise, the cement should provide an appropriate working time for the clinician to apply the material to the patient’s mouth. To fulfill this condition, the DC should be maintained at a low level for several minutes after the materials are mixed. By estimating the working time from the time point at which each cement showed significant changes, G6 had the shortest working time of 2 min, whereas G1 had the longest working time of approximately 20 min. Considering that the working time of commercial dual-cured resin cements ranges from 2 to 5 min [20], G3, 5, and 6 seem to have an appropriate working time.

Unreacted monomers of the resin cement have a cytotoxic effect on cells in the surrounding tissue [33]. A previous study showed that resin cement with a lower DC caused higher cytotoxicity due to the presence of more unreactive monomers [33,34]. The results of this study are consistent with those of previous studies. Cell viability 10 min after mixing followed the relationship between the DC and cytotoxicity. Groups with a higher DC at 10 min (G3, 5, and 6) had higher cell viability than the groups with a lower DC (G1, 2, and 4). In previous studies, self-adhesive resin cements showed significantly higher cytotoxicity at a low DC because of their acidic components used for etching teeth [11,30]. The large differences in cell viability between the higher DC group and the lower DC group at 10 min after the mixing of viability at 10 min confirm the previous results. This result suggests that achieving a higher DC in the early stage of the restoration setting is necessary for self-adhesive resin cements to avoid unfavorable biological results caused by unreacted components. As shown by the DC results in the light-curing mode, achieving a higher DC through photo-polymerization in the early stages is possible. The results show that the light-curing method of radiating 1000 mW/cm^2^ light intensity for 30 s is appropriate for the photo-polymerization of dual-resin cement under highly translucent zirconia with a 1 mm thickness. However, in the case of setting an indirect restoration with a thicker attenuating ceramic, light curing with a higher light intensity for a longer duration may be necessary to derive an appropriate photo-polymerization [32]. Continuous irradiation with high-intensity light for longer durations causes thermal changes in the restoration, which can cause irreversible pulp damage [32]. Providing a pause between irradiations may be necessary to avoid unfavorable results. When using metal or bilayer zirconia, which completely blocks the curing light for self-adhesive resin cement, resin cement with faster self-curing is safer because of the hazard from unreacted components.

The film thickness and flow properties are related to parameters such as viscosity, filler size, the filler/matrix ratio, and temperature [35,36,37]. A larger filler size and a higher filler/matrix ratio result in a higher film thickness [38,39]. Storing at a lower temperature increases the fluidity of resin cement [19]. Resin cements with higher viscosities are thicker and have lower flowability [38,40]. In this study, the filler size, filler/matrix ratio, and temperature were similar among all groups. The differences in the DC among the groups did not affect the film thickness or flow properties in this study. Similar results were obtained in this study compared to previous results, where the film thickness and flowability were inversely proportional [41,42]. Although a thinner film thickness and higher flowability were observed in G1 and 4, which commonly contained fewer self-curing initiators or self-curing activators than the other groups, future experiments are necessary to determine the effect of each component on film thickness and flowability. As the film thickness of all groups was below 50 μm, all groups satisfied the reference of ISO standard 4049 for polymer-based restorative materials and were acceptable for clinical use.

In this study, various experimental groups were created and utilized, while maintaining other components, excluding those that influenced self-curing. Therefore, it is advantageous to accurately evaluate the changes in the properties of resin cement due to the self-curing initiator, self-curing activator, and accelerator. However, a limitation of this study is that the experiments were conducted under conditions different from those of the patient’s oral cavity, including temperature and humidity, thus not fully reproducing the environment in which resin cement is actually used. In particular, resin cement can be influenced by temperature variations in terms of the DC [43,44]. However, the FT-NIR instrument used to measure the DC is recommended for use at room temperature; hence, such differences inevitably occur. To overcome these limitations, a comprehensive approach that considers various variables and conditions is required.

## 5. Conclusions

Within the limitations of this study, the following conclusions were drawn. All supplemental components had an effect on increasing the DC, but a greater increase in the DC was observed in the following order: activator > accelerator > initiator. The cytotoxicity of the cements was related to the DC values, as cements with lower DC values exhibited higher cytotoxicity. The film thickness met the ISO 4049 standard for all the groups. Utilizing an activator is the most effective approach for enhancing the DC in self-adhesive resin cement.

## Figures and Tables

**Figure 1 materials-17-03572-f001:**
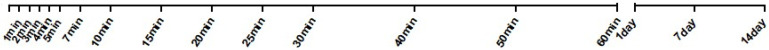
The timeline for the DC measurement.

**Figure 2 materials-17-03572-f002:**
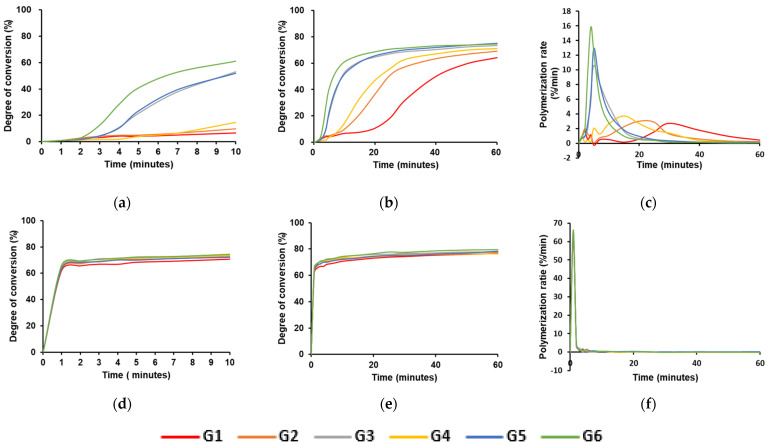
The results of the DC and the polymerization rate. (**a**) The changes of the DC without light curing during 10 min, (**b**) the changes of the DC without light curing during 60 min, (**c**) the polymerization rate per minute as the time flow without light curing, (**d**) the changes of the DC after light curing during 10 min, (**e**) the changes of the DC after light curing during 60 min, and (**f**) the polymerization rate per minute as the time flow after light curing.

**Figure 3 materials-17-03572-f003:**
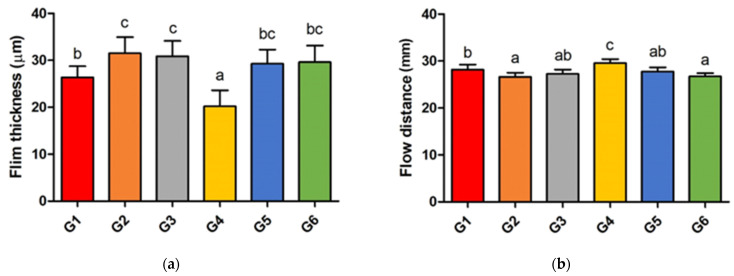
(**a**) The film thickness and (**b**) flow distance of self-adhesive resin cements. Significant differences are indicated by different letters.

**Figure 4 materials-17-03572-f004:**
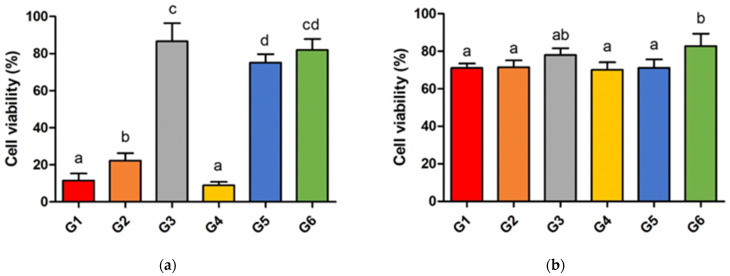
(**a**) Cell viability of self-adhesive resin cements after self-curing for 10 min and (**b**) cell viability of self-adhesive resin cements after self-curing for 60 min. Significant differences are indicated by different letters.

**Figure 5 materials-17-03572-f005:**
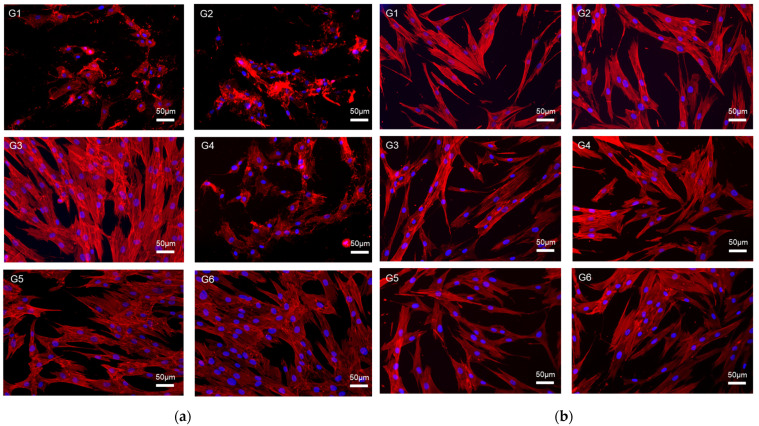
(**a**) Cell morphology of self-adhesive resin cements after self-curing for 10 min and (**b**) cell morphology of self-adhesive resin cements after self-curing for 60 min. F-actin was stained with red fluorescent dye and nucleic was stained with blue fluorescent dye.

**Figure 6 materials-17-03572-f006:**
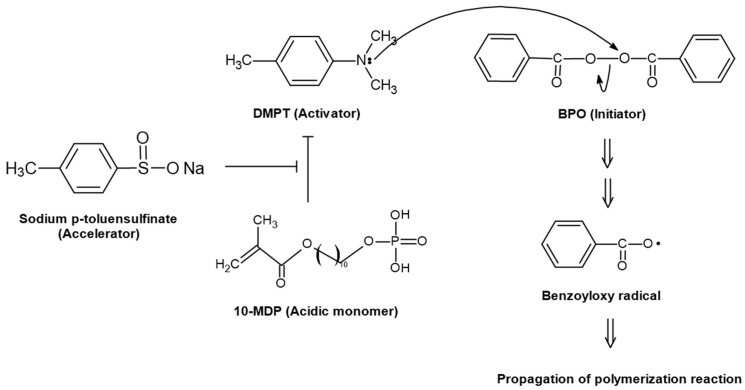
Reaction mechanism of the initiator, activator, and accelerator.

**Table 1 materials-17-03572-t001:** Components of experimental self-adhesive resin cements.

	Components
Paste A	TEGDMA, UDMA, 10-MDP, BPO, TP, barium glass filler
Paste B	TEGDMA, UDMA, TP, CQ, EDMAB, DMPT, SPTS, barium glass filler

**Table 2 materials-17-03572-t002:** The concentration of the components of self-adhesive resin cements used in this study.

	G1	G2	G3	G4	G5	G6
BPO (self-curing initiator)	1	2	1	1	2	1
DMTP (self-curing activator)	1	1	2	1	1	2
SPTS (accelerator)				1	1	1

**Table 3 materials-17-03572-t003:** Statistical analysis of the DC (%) in self-curing mode. The data show the mean and standard deviation.

	Group	G1	G2	G3	G4	G5	G6
Time							
1 min		0.91 (0.68) ^A,a^	0.84 (0.50) ^A,a^	0.93 (1.08) ^A,a^	0.95 (0.49) ^A,a^	0.69 (0.61) ^A,a^	0.93 (0.77) ^A,a^
2 min		2.65 (1.53) ^A,a^	2.85 (1.47) ^A,a^	1.91 (1.10) ^A,a^	1.16 (0.58) ^A,a^	1.61 (0.73) ^A,a^	2.77 (2.24) ^AB,a^
3 min		3.18 (1.43) ^A,a^	4.14 (1.29) ^A,a^	3.96 (2.37) ^A,a^	1.61 (0.88) ^A,a^	4.46 (2.71) ^A,a^	12.61 (8.32) ^B,a^
4 min		4.34 (0.81) ^A,a^	4.95 (1.31) ^A,a^	10.90 (5.52) ^A,a^	1.84 (1.05) ^A,a^	10.41 (7.70) ^A,a^	28.44 (11.26) ^C,b^
5 min		4.11 (1.30) ^A,a^	5.04 (1.15) ^A,a^	21.53 (10.55) ^B,b^	3.92 (2.01) ^A,a^	23.33 (11.39) ^B,b^	40.81 (10.64) ^D,c^
7 min		5.15 (0.59) ^A,a^	6.58 (1.36) ^A,a^	37.73 (10.33) ^C,b^	6.48 (3.15) ^AB,a^	39.34 (13.71) ^C,b^	52.72 (7.93) ^E,c^
10 min		6.75 (1.49) ^A,a^	9.91 (2.50) ^A,a^	53.01 (4.90) ^D,b^	14.55 (4.83) ^B,a^	51.79 (8.02) ^D,b^	61.02 (4.41) ^EF,b^
15 min		7.64 (1.68) ^A,a^	21.36 (9.68) ^B,b^	61.23 (2.49) ^DE,d^	33.38 (5.79) ^C,c^	60.89 (5.43) ^DE,d^	65.98 (2.56) ^FG,d^
20 min		10.71 (2.55) ^AB,a^	36.53 (10.17) ^C,b^	64.64 (2.10) ^EF,d^	46.97 (10.56) ^D,bc^	65.74 (3.14) ^EF,d^	68.62 (1.65) ^FG,d^
25 min		18.59 (8.23) ^B,a^	51.06 (6.38) ^D,b^	67.12 (2.19) ^EF,d^	55.93 (7.48) ^DE,bc^	68.39 (2.83) ^EF,d^	70.67 (1.32) ^FGH,d^
30 min		32.25 (15.35) ^C,a^	57.47 (4.09) ^DE,b^	68.75 (1.54) ^EF,c^	63.01 (2.79) ^EF,bc^	70.22 (2.01) ^EF,c^	71.51 (1.29) ^GH,c^
40 min		50.13 (11.54) ^D,a^	63.55 (2.07) ^EF,b^	70.37 (1.76) ^EFG,b^	67.08 (1.86) ^F,b^	72.04 (1.59) ^F,b^	73.13 (0.83) ^GH,b^
50 min		59.66 (4.28) ^DE,a^	66.86 (2.02) ^EF,a^	72.34 (1.30) ^FG,b^	69.96 (1.25) ^F,a^	73.76 (1.17) ^FG,b^	73.70 (0.99) ^GHI,b^
60 min		64.33 (3.37) ^E,a^	69.19 (1.78) ^F,a^	73.40 (1.43) ^FG,a^	71.11 (2.00) ^FG,a^	75.22 (1.75) ^FGH,a^	74.52 (0.58) ^GHI,a^
1 day		80.35 (1.38) ^F,a^	82.73 (1.45) ^G,a^	80.63 (0.69) ^GH,a^	80.71 (0.75) ^GH,a^	82.96 (0.79) ^GHI,a^	80.76 (0.92) ^HIJ,a^
1 week		83.45 (0.54) ^F,a^	85.52 (1.91) ^G,a^	83.76 (0.64) ^H,a^	83.74 (0.33) ^H,a^	85.38 (0.66) ^HI,a^	83.91 (0.63) ^IJ,a^
2 weeks		84.08 (0.73) ^F,a^	86.53 (0.30) ^G,a^	84.77 (0.46) ^H,a^	84.77 (0.53) ^H,a^	86.10 (0.38) ^I,a^	84.83 (0.44) ^J,a^

Significant differences are indicated by different letters (uppercase letters within columns and lowercase letters within rows).

**Table 4 materials-17-03572-t004:** Statistical analysis of the DC (%) in light-curing mode. The data show the mean and standard deviation.

	Group	Group1	Group2	Group3	Group4	Group5	Group6
Time							
1 min		62.30 (1.79) ^A,a^	63.59 (1.72) ^A,ab^	65.80 (1.59) ^A,b^	64.57 (2.31) ^A,ab^	64.12 (3.06) ^A,ab^	66.31 (1.08) ^A,b^
2 min		65.56 (0.80) ^AB,a^	67.31 (1.32) ^AB,ab^	68.91 (0.79) ^AB,ab^	68.07 (2.15) ^AB,ab^	68.21 (2.51) ^B,ab^	69.36 (1.33) ^AB,b^
3 min		66.79 (0.86) ^B,a^	69.28 (1.19) ^BC,ab^	71.13 (1.25) ^BC,b^	68.94 (1.69) ^B,ab^	68.83 (2.47) ^BC,ab^	70.46 (1.56) ^B,b^
4 min		66.75 (1.13) ^B,a^	70.58 (1.55) ^BCD,b^	71.21 (1.30) ^BC,b^	70.97 (1.96) ^BC,b^	70.18 (3.41) ^BCD,b^	71.46 (2.02) ^BC,b^
5 min		68.26 (1.01) ^BC,a^	70.65 (2.00) ^BCD,ab^	72.52 (1.45) ^BCD,b^	71.52 (2.01) ^BCD,ab^	70.04 (3.20) ^BCD,ab^	72.04 (2.13) ^BCD,b^
7 min		69.07 (1.52) ^BC,a^	71.20 (2.11) ^CDE,ab^	72.88 (0.53) ^CD,b^	72.13 (1.68) ^CD,ab^	71.26 (2.79) ^BCDE,ab^	72.63 (1.80) ^BCD,b^
10 min		70.60 (1.83) ^CD,a^	72.77 (2.32) ^CDEF,ab^	74.22 (0.97) ^CDE,b^	73.48 (2.78) ^CDE,ab^	72.21 (3.22) ^CDEF,ab^	74.38 (2.20) ^CDE,b^
15 min		71.81 (1.54) ^CDE,a^	73.73 (2.05) ^DEFG,ab^	75.27 (0.79) ^DEF,b^	73.36 (2.60) ^CDE,ab^	73.17 (3.26) ^DEFG,ab^	75.34 (1.76) ^DEF,b^
20 min		73.01 (1.81) ^DEF,a^	74.71 (1.89) ^EFGH,ab^	75.93 (0.49) ^DEF,ab^	74.14 (2.20) ^CDE,ab^	74.59 (2.71) ^EFGH,ab^	76.51 (1.63) ^EFG,b^
25 min		73.78 (1.88) ^DEF,a^	75.29 (1.81) ^FGH,ab^	76.06 (0.77) ^DEF,ab^	74.93 (2.50) ^DE,ab^	75.04 (2.15) ^FGHI,ab^	77.80 (1.43) ^EFG,b^
30 min		74.20 (1.91) ^DEF,a^	76.22 (0.99) ^FGH,a^	76.72 (0.72) ^EF,a^	75.19 (2.65) ^DE,a^	75.15 (2.46) ^FGHI,a^	77.56 (1.15) ^EFG,a^
40 min		75.27 (1.43) ^EF,a^	77.15 (1.47) ^GH,a^	76.96 (0.88) ^EF,a^	75.93 (1.97) ^E,a^	76.31 (1.80) ^GHI,a^	78.66 (1.15) ^FG,a^
50 min		75.89 (1.37) ^F,a^	77.86 (1.20) ^H,ab^	77.41 (0.56) ^EF,ab^	76.29 (1.98) ^E,ab^	77.01 (1.19) ^HI,ab^	79.31 (1.08) ^G,b^
60 min		76.59 (1.75) ^F,a^	77.59 (1.38) ^H,a^	78.15 (1.00) ^F,a^	76.43 (2.36) ^E,a^	78.54 (0.78) ^I,a^	79.59 (0.75) ^G,a^
1 day		83.82 (0.51) ^G,a^	84.65 (1.05) ^I,a^	84.26 (0.55) ^G,a^	83.15 (1.29) ^F,a^	84.93 (0.77) ^J,a^	84.35 (0.65) ^H,a^
1 week		85.44 (0.75) ^G,a^	86.32 (1.08) ^I,a^	86.67 (0.50) ^G,a^	85.57 (1.10) ^F,a^	87.07 (0.49) ^J,a^	87.04 (0.39) ^H,a^
2 weeks		86.70 (0.84) ^G,a^	87.76 (0.69) ^I,a^	87.29 (0.68) ^G,a^	86.57 (0.94) ^F,a^	87.49 (0.41) ^J,a^	87.61 (0.78) ^H,a^

Significant differences are indicated by different letters (uppercase letters within columns and lowercase letters within rows).

## Data Availability

The original contributions presented in the study are included in the article, further inquiries can be directed to the corresponding authors.

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
