# Peer review of "The Effect of the Initiator/Activator/Accelerator Ratio on the Degree of Conversion, Film Thickness, Flow, and Cytotoxicity of Dual-Cured Self-Adhesive Resin Cements"

_materials, 2024, doi:10.3390/ma17143572_

Round 1
Reviewer 1 Report
Comments and Suggestions for Authors
Overall a very good research study
line 72-73 improper seating of definitive restoration causes many problems as you mentioned. But it might cause occlusal interference requiring minor or major selective grinding of the prosthesis which might be a reason for a repetition or new manufacturing of the prosthesis in some cases.The low film thickness of RCs which is significantly lower than pre-heated photo-cured resin cements that require a better level of expertise to achieve a clinically acceptable film thickness.(DOI: 10.1111/jerd.12988)
Line 321-325 Film thickness of the measured RC in all groups is similar to most RC in market. It is known that conventional Glass ionomer or even RMGI cements have even lower film thickness and film thickness in RC is influenced by size or addition of fillers even nanoparticles (10.3390/ijms24032067 )
Reviewer 2 Report
Comments and Suggestions for Authors
Please complete the legend for figure #2. I understand that the reader can identify the correct designation of a given curve based on the same color in another image. But that's impractical. It is appropriate to place the legend directly in the image or near the image.
In Table No. 3 and 4, the units of the values ​​are not given. Double values ​​(base and bracketed) are not explained here. Please explain the values ​​and indicate the units near the tables.
I would recommend to consider naming chapter 3 "results" in a different way.
As the main contribution, I appreciate the quantification of the benefits of individual modifications and in determining their order of contribution. Thank you for your contribution to this topic.
Reviewer 3 Report
Comments and Suggestions for Authors
This paper addresses an interesting subject which is to investigate the influence of the activator, initiator, and accelerator ratio on some physical and biological properties. The manuscript is well-written, and the method supports the results and conclusions. Also, the manuscript is well structured, giving to the readers a pleasant reading. The discussion pointed to previous studies and was able to provide reasonable explanations for all the results. Also, the authors pointed out the limitations of the study.
However, some considerations are necessary and are related below:
1. According to the authors “this study aimed to evaluate the effects of the initiator/activator/accelerator ratio” on some physical and biological properties. CQ/EDMAB is also an initiator, however its concentration was not modified for any group. The authors should emphasize that only self-curing initiator and activator, BPO and DMPT were investigated. Also, it would be useful to show the concentrations for each component of the resin cement.
2. Regarding film thickness and flow ability, it is expected that the chemical cure had already started at 60 seconds and 180 seconds, respectively. Is this taken into account for the tests¿
3. Regarding cell viability, it is not clear why the authors did not investigate photo-cured specimens. Also, the authors did not inform the n for the test.
4. In Figure 2, it is not possible to identify the curve for each group, impairing the comprehension of the results.
5. In Figure 3, it is not possible to identify the mean and standard deviation for each group. Or the standard deviation could be provided in the Results (text).
